# Optimization of the Design Configuration and Operation Strategy of Single-Pass Seawater Reverse Osmosis

**DOI:** 10.3390/membranes12111145

**Published:** 2022-11-15

**Authors:** Seung Ji Lim, Seo Jin Ki, Jae-Lim Lim, Kyunghyuk Lee, Jihye Kim, Jeongwoo Moon, Joon Ha Kim

**Affiliations:** 1School of Earth Sciences and Environmental Engineering, Gwangju Institute of Science and Technology (GIST), Gwangju 61005, Republic of Korea; 2Center for Water Cycle Research, Korea Institute of Science and Technology (KIST), Seoul 02792, Republic of Korea; 3Department of Environmental Engineering, Gyeongnam National University of Science and Technology, 33 Dongjin-ro, Jinju 52725, Republic of Korea; 4Korea Water Resources Corporation (K-Water), Daejeon 34045, Republic of Korea

**Keywords:** reverse osmosis process, process optimization, numerical study, design configuration, operating strategy

## Abstract

The numerical study was conducted to compare process performance depending on the pump type and process configuration. The daily monitoring data of seawater temperature and salinity offshore from Daesan, Republic of Korea was used to reflect the site-specific seawater conditions. An algorithm for reverse osmosis in constant permeate mode was developed to simulate the process in time-variant conditions. Two types of pumps with different maximum leachable efficiencies were employed to organize pump-train configuration: separated feed lines and common pressure center design. The results showed pump type and design configuration did not have a significant effect on process performance. The annual means of specific energy consumption (*SEC*) for every design configuration were under 2 kWh/m^3^, except for a worst-case. The worst-case was decided when the pump was operated out of the best operation range. The two operation strategies were evaluated to determine the optimal configuration. The permeate flow rate was reduced to 80% of the designed permeate flow rate with two approaches: feed flow rate reduction in every train and pump shutdown in a specific train. The operation mode with feed flow rate reduction was more efficient than the other. The operating pressure reduction led to a decrease in *SEC*.

## 1. Introduction

Seawater reverse osmosis (SWRO) has matured into the leading technology in the desalination market [1,2]. There are two main reasons for this: (1) reduced specific energy consumption (*SEC*) thanks to advances in membrane technology and (2) the development of energy recovery devices (ERD) [3,4]. Water permeability and salt selectivity have been improved in the last few decades. Increased productivity at the membrane level reduces the *SEC* [1,5,6]. The development of ERDs has contributed to reducing the *SEC* at the process level. ERDs allow the recovery and reuse of pressure energy discarded through the concentrate flow, reducing the electric power consumed by the high-pressure (HP) pump [7,8,9,10]. However, although the efficiency of SWRO has been significantly improved since the technology was first developed, it is still an energy-intensive process [5,7,11,12]. The reverse osmosis (RO) unit accounts for 80% of the energy consumption in the entire SWRO process [4]. Many studies regarding the reduction of the intensive energy usage in SWRO process have been conducted to optimize the processes or to employ renewable energy sources [13,14,15]. Ruiz-García and Nuez [16] conducted an assessment of safe operating windows and optimization of SWRO using process model. The results showed that the maximum difference of *SEC* depending on the types of membrane modules was 0.24 kWh m^−3^ in the optimal operating points. Takabatake et al. [17] provided comprehensive reviews on the stabilization and performance of the advanced SWRO technology. However, there are still limited studies that consider the optimization of HP pump configuration.

One way to reduce energy usage in the RO process is to optimization of HP pump operation, which consumes most of the energy used in the RO unit. There are two practical methods to maintain pump efficiency near the maximum leachable efficiency. One method is to employ a variable frequency drive (VFD) in the HP pump, which prevents throttling losses due to the use of a throttling valve [4,5,8]. It means that the operating pressure to produce the designed permeate flow can be efficiently adjusted under conditions of feed water temperature and salinity fluctuations [18]. The other method is to use a large-capacity pump. The pump’s efficiency tends to increase as its capacity increases [18,19,20]. Therefore, pump efficiency could be changed according to the pump-train configuration. There are two types of design configuration: pump-train configuration: separated feed lines and common pressure center design [18,21]. One high-pressure pump supplies the feed water to a specific train in pump-train configuration with the separated feed lines design. On the other hand, the different number of pumps and trains are utilized to configure common pressure center design. Several pumps are coupled to form a common pressure center. The common pressure center feeds multiple trains. The pump for common pressure center design usually has a higher capacity and efficiency than the pump for separated feed lines. The Ashkelon SWRO plant is one of the most famous plants using the common pressure center design [11,22,23]. Using a large-capacity HP pump, they were able to achieve an HP pump efficiency of 88.5% [11,18,22,23]. Even though the Ashkelon plant demonstrated the advantages of a large-capacity pump and the common pressure center design, the optimal design configuration for single-pass SWRO has not been fully discussed. Although several studies explored the relation between pump capacity and pump efficiency, they only investigated the maximum reachable efficiency in relation to the pump’s capacity [18,21]. A comparison of two different pump configurations was briefly conducted in one study based on the pump’s specifications [18]. Many studies have addressed the advantages of a large-capacity pump and the common pressure center design with few numerical studies and the case of *SEC* reduction in the Ashkelon SWRO plant [1,4,5,6,24]. Therefore, research into the optimization of design and operation considering the RO configuration details is in high demand.

In this study, we conducted a RO simulation according to design configuration and operation strategy, considering site-specific seawater temperature and salinity. The target area was Daesan, one of the biggest industrial complexes in the Republic of Korea. Monitoring data spanning a one-year period (2013) was employed. An algorithm for the RO process in constant permeate mode was developed using a previously developed RO model [23,25]. A time-variant pump efficiency equation was used to test the influence of the pump’s efficiency on the RO process. A numerical simulation based on six design configurations with different pump types and design configurations was conducted to determine the optimal design with the lowest possible *SEC*. Finally, the best operation strategy in terms of permeate flow reduction was investigated. The specific objectives of this research were (1) to propose an algorithm that can simulate the RO process in constant permeate mode under time-variant conditions, (2) to compare the process performance between different process unit configurations (pump types and numbers of pumps, trains, and pressure vessels [PVs]), and (3) to evaluate the effect of the operating modes on process performance.

## 2. Materials and Methods

### 2.1. Seasonal Variations of Seawater Temperature and Salinity in Daesan

Figure 1 shows the daily data of seawater temperature and total dissolved solids (*TDS*) concentration offshore from Daesan during the measurement period. The data is open access, published by the Korea Hydrographic and Oceanographic Agency [26]. The daily mean of hourly monitoring data was calculated for the daily simulation. An interpolation algorithm was used to make up for missing data due to the malfunction of measurement devices. The seawater temperature in this region is subject to seasonal variations. Its minimum value (4 °C) was recorded on day 51 of the year, after which it continuously increased, peaking on day 243 (26 °C). The average temperature was 13 °C. In contrast, the *TDS* concentration did not show seasonal trends, fluctuating randomly between 24,270 and 31,040 mg/L. These data were used in the scenario study to simulate on-site seawater conditions.

### 2.2. Model of Reverse Osmosis in Constant Permeate Mode

One way to numerically describe the transport in the spiral-wound membrane is to use the solution-diffusion model. This model has been used in several studies to calculate the performance of the RO process only in constant pressure mode [25,27,28]. In this study, we developed an improved algorithm to simulate the RO process of a fixed amount of permeate water by manipulating the operating pressure.

#### 2.2.1. Numerical Model for the Spiral-Wound Membrane

The transport phenomena of the solution in the membrane are described by two types of flux: water and salt flux. The following equation was used to calculate water flux *v_w_* [27]:(1)$$v_{w}\left(x,t\right)=\frac{\Delta p\left(x,t\right)-\Delta \pi \left(x,t\right)}{\mu \left(t\right)\cdot R_{t}\left(x,t\right)\cdot TCF_{A}\left(t\right)}$$
where Δ*p* is the trans-membrane pressure (TMP). To take into account seasonal feedwater variations, the variables in the water flux equations include temperature and *TDS* concentration. The varying osmotic pressure Δ*π* was calculated as follows [27]:(2)$$\Delta \pi \left(x,t\right)=\left(23745+64.784\cdot c_{w}\left(x,t\right)+1.7753\times 10^{-4}\cdot c_{m}^{2}\left(x,t\right)\right)\times \frac{T_{f}\left(t\right)}{298.15}$$
where *T_f_* and *c_m_* indicate the feedwater temperature and salt concentration on the membrane wall, respectively. An empirical equation was used to describe changes in seawater viscosity *μ* at various temperatures [29]:(3)$$\mu \left(t\right)=2.414\times 10^{-5}\times 10^{\frac{247.8}{T_{f}\left(t\right)-140}\;}$$
where *R_t_* is the total membrane resistance due to foulant accumulation on the membrane. It is calculated by adding the intrinsic membrane resistance *R_m_* and the resistance of membrane fouling *R_a_. R_a_* is affected by the amount of permeate filtrated by the membrane [25]:(4)$$R_{t}\left(x,t\right)=R_{m}+R_{a}\left(x,t\right)$$
(5)$$R_{a}\left(x,t\right)=k_{fp}\int_{0}^{t}v_{w}\left(x,\tau \right)d\tau \;$$
where *k_fp_* denotes the fouling potential in feedwater and *τ* is a dummy variable for integration with respect to time. The temperature correction factor *TCF* is adopted to modify *R_t_* during the simulation [30]:(6)$$TCF_{A}\left(t\right)=exp\left(3000\left(\frac{1}{T_{f}\left(t\right)}-\frac{1}{T_{ref}}\right)\right)$$
where *T_ref_* is the reference temperature of 298.15 K. The drop in TMP along the membrane channel was calculated by the following equation [25,31]:(7)$$\Delta p\left(x,t\right)=\Delta p_{0}\left(t\right)-\frac{12\cdot k_{fr}\cdot \mu \left(t\right)}{H^{2}}\int_{0}^{x}u\left(\xi ,t\right)d\xi$$
where *k_fr_* is the friction coefficient in the membrane channel, *ξ* is a dummy variable for integration with respect to distance, and Δ*p*_0_ is the TMP at the entrance of the membrane channel. It is noted that Δ*p*_0_ varies with time to calculate the TMP required to produce a constant amount of permeate water. The equation for the feed flow rate *u* is derived from the mass conservation law in the membrane channel [25]:(8)$$u\left(x,t\right)=u_{0}-\frac{1}{H}\int_{0}^{x}v\left(\xi ,\;t\right)d\xi$$
where *H* is the height of the membrane channel. The salt flux vs. was calculated by the following equation [30,31]:(9)$$v_{s}\left(x,t\right)=B\cdot TCF_{B}\left(t\right)\cdot c\left(x,t\right)$$
(10)$$TCF_{B}\left(t\right)=exp\left(-4500\left(\frac{1}{T_{f}\left(t\right)}-\frac{1}{T_{ref}}\right)\right)$$
where *B* denotes the salt permeability coefficient of the membrane. The *TCF* for *B* (*TCF_B_*) is also used to reflect the fluctuation of the feedwater temperature, and *c* is the salt concentration in the bulk flow, which was expressed as follows [27]:(11)$$c\left(x,t\right)=\frac{1}{u\left(x,t\right)}\left[c_{0}\left(t\right)u_{0}-\frac{B\cdot TCF_{B}\left(t\right)}{H}\int_{0}^{x}c\left(\xi ,t\right)d\xi \right]$$
where *c*_0_ is the *TDS* concentration at the entrance of the membrane channel, which is equal to the varying *TDS* concentration in the feedwater. The concentration polarization was calculated using the wall concentration *c_m_* [32]:(12)$$c_{m}\left(x,t\right)=c_{0}\left(t\right)+e^{-\frac{v_{w}\left(x,t\right)\cdot H}{D}}\frac{r\left(x,t\right)\cdot c_{0}\left(t\right)}{u\left(x,t\right)\cdot H}\int_{0}^{x}v_{w}\left(\xi ,t\right)d\xi +\frac{r\left(x,t\right)\cdot c_{0}\left(t\right)\cdot v_{w}\left(x,t\right)}{u\left(x,t\right)\cdot H}\int_{0}^{x}v_{w}\left(\xi ,t\right)d\xi$$
where *D* is the hydraulic dispersion coefficient and *r* is the rejection rate of the membrane, which was calculated as follows [27]:(13)$$r\left(x,t\right)=1-B\cdot \frac{TCF_{B}\left(t\right)}{v_{w}\left(x,t\right)}$$

It is noted that the system boundary of the model described in this section is an element of the spiral-wound membrane. The system boundary is extended to the RO system in Section 2.2.2. The parameters used in this study are presented in Table 1.

#### 2.2.2. Performance of the RO Model

The process performance was evaluated using the permeate flow rate *Q_p_*, permeate *TDS* concentration *C_p_*, and specific energy consumption *SEC*. The *Q_p_* of the RO system was calculated as follows: (14)$$Q_{p}\left(t\right)=N_{TRN}\cdot N_{PV}\cdot W\cdot \int_{0}^{TL}v_{w}\left(x,t\right)dx$$
(15)$$Q_{s}\left(t\right)=N_{TRN}\cdot N_{PV}\cdot W\cdot \int_{0}^{TL}v_{s}\left(x,\;t\right)dx$$
where *W* and *TL* are the width and total length of the membrane elements, respectively. The permeate flow rate in the RO system can be determined by multiplying the number of trains in an RO system *N_TRN_* by the number of pressure vessels in a train *N_PV_*. A similar equation is used to calculate the salt flow rate through the membrane *Q_s_. C_p_* was determined using the *Q_p_* and *Q_s_* as follows:(16)$$C_{p}\left(t\right)=\frac{Q_{s}\left(t\right)}{Q_{p}\left(t\right)}$$

The *SEC* of the RO system was calculated as the ratio of consumed energy to the unit volume of produced permeate water [33]:(17)$$SEC\left(t\right)=\frac{W_{HP}\left(t\right)\cdot N_{HP}+W_{BP}^{ERD}\left(t\right)\cdot N_{BP}}{Q_{p}\left(t\right)}$$
where *W_HP_* and $W_{BP}^{ERD}$ are the work done by each HP pump and the booster pump after the ERD, respectively. *W_HP_* is proportional to the pressure increased by the pump and the flow rate through the pump *Q_fp_*, which was calculated as follows: (18)$$W_{HP}\left(t\right)=\left(p_{f}\left(t\right)-p_{in}^{HP}\right)\cdot \frac{Q_{fp}}{\eta_{HP}\cdot \eta_{VFD}}$$
where *p_f_* and $p_{in}^{HP}$ are the operating pressure applied to the feed flow rate and the feed pressure at the entrance of the HP pump, respectively, and *η_HP_* and *η_VFD_* are the efficiencies of the HP pump and variable frequency drive. The efficiency of the HP pump *η_HP_* was expressed as an equation for *p_f_* and *Q_fp_*, which is introduced in Section 2.2.3. In this study, it was assumed that *η_VFD_* has a fixed value during process operation. $W_{BP}^{ERD}$ was determined as follows:(19)$$W_{BP}^{ERD}\left(t\right)=\left(p_{f}\left(t\right)-p_{in}^{ERD}-\eta_{ERD}\left(p_{b}\left(t\right)-p_{out}\right)\right)\cdot \frac{Q_{b}\left(t\right)}{N_{BP}\cdot \eta_{BP}}$$
where $p_{in}^{ERD}$, *p_b_*, and *p_out_* are the feed flow pressure at the entrance of the ERD, the brine pressure, and the pressure from the ERD, respectively, *Q_b_* is the brine flow rate in the RO system, and *η_BP_* is the booster pump’s efficiency after the ERD. It is noted that *x* and *t* are used to indicate the position in the membrane channel and operation time, respectively, while *i* and *j* are used to describe the position and time in the numerical algorithm. It is noted that the VFP and boost pump efficiencies were set to constant values in this study to evaluate the effect of HP pump efficiency.

#### 2.2.3. Efficiency Equation for the HP Pump

The HP pump employed in this study was a centrifugal pump. The specifications provided by the pump manufacturer contain the pump performance curve. The efficiency equation was derived using these specifications. First, the flow rate through the HP pump was determined as follows: (20)$$Q_{fp}\left(t\right)=Q_{f}\left(t\right)\cdot \frac{Rec}{N_{HP}}$$
where *Q_f_* is the rate of feed flow to the RO system. The multiplication of *N_HP_* over the recovery ratio *Rec* means that *Q_fp_* is the rate of flow distributed to each HP pump. In this study, a portion of feed flow equal to the brine flow was pressurized in the ERD. *Q_fp_* was used to calculate *η_HP_*, which was determined using the pump datasheet from a confidential pump manufacturer. *η_HP_* was calculated using the following form of equation: (21)$$\;\eta_{HP}=p_{1}+p_{2}\cdot Q_{fp}\left(t\right)+p_{3}\cdot p_{f}\left(t\right)+p_{4}\cdot Q_{fp}^{2}\left(t\right)+p_{5}\cdot Q_{fp}\left(t\right)\cdot p_{f}+p_{6}\cdot p_{f}^{2}\left(t\right)$$
where the coefficients *p*_1_ to *p*_6_ are the fitting parameters from a curve-fitting method. We employed two types of high-pressure pumps, which have different coefficients *p*. The coefficients for each pump type are shown in Table 2. The efficiency of the HP pump *η_HP_* is a function of two variables, *Q_fp_* and *p_f_*. Consequently, *η_pump_* has an efficiency surface varying with the rate of flow to the HP pump and the operating pressure. The type 2 pump has a larger capacity and higher maximum efficiency than the type 1 pump.

#### 2.2.4. Algorithm for RO in Constant Permeate Mode

Figure 2 illustrates an algorithm for RO in constant permeate mode. The data mentioned in Section 2.1 was imported as feedwater temperature and *TDS* concentration. The daily data was delivered to the RO model in every iteration. All equations used in the RO model are described in Section 2.2.1. The flow chart in the pressure calculation part calculates the transmembrane and operating pressure of the RO model. The algorithm calculates the permeate flow rate and compares it with the target permeate flow rate *Q_pt_*. If the deviation between *Q_pt_* and calculated permeate flow rate exceeds the margin of flow rate *Q_m_*, the algorithm modifies the operating pressure. The algorithm repeats this iteration until the calculated permeate flow rate meets the target permeate flow rate. The permeate flow rate and operation pressure determined in this part are used to calculate the performance of the RO model. The algorithm is set to end the simulation when the simulated time reaches the operation time.

### 2.3. Optimization of Design and Operation of RO in Permeate Mode

Table 3 shows the conditions for the scenario study of the design and operation of the RO process (see Appendix A for a graphical summary). The process performance of each configuration design was compared to the others. The pump type and arrangement and the number of trains were changed in the fixed permeate flow rate (60,000 m^3^/d). Six design configuration ratios were established for the pump and the train arrangement to avoid the repetition of simulation. They consisted of a pump-train configuration with separated feed lines or common pressure center design. The number of PVs was decided in the range between 100 to 300 in increments of 5. The constraint for the maximum flux in the PVs was set to 32.313 LMH. The pump operation ranges of 16,800–26,880 and 9600–14,400 m^3^/d were used as constraints for the type 1 and the type 2 pump, respectively. In the RO operation scenario, we considered a situation that reduces productivity during the operation. The permeate flow rate was changed from 60,000 to 48,000 m^3^/d. Two different ways of operation were adopted. The first strategy was to reduce the flow rate in every PV to 80% (operation strategy 1). The second strategy was to shut down the pump or lock the valve before the RO train (operation strategy 2). It is noted that the number of PVs per train in the operation step was set according to the minimum *SEC* in each case.

### 2.4. Statistical Analysis to Compare the Process Performance

The process performance was compared using statistical analysis. This section focuses on the introduction of the statistical method to understand and analyze the simulation results because it is out of scope for this study to explain each statistical analysis’s detail. Analysis of variance (ANOVA) is a statistical test to evaluate the difference of mean among two or more groups. *F*-value is employed to estimate the difference as follows [34]:(22)$$F=\frac{between\;groups\;variance}{within\;groups\;variance}$$

The normality and homogeneity of variance should be evaluated to conduct ANOVA. It is noted that the normality of data was assumed in this study. Levene’s test is employed to check homogeneity of variance. As you can see from the definition of F-value, the ANOVA test can only reveal whether there are differences among the groups. Therefore, the post hoc analysis is required to find which group is different from the others. Student-Newman-Keuls (SNK) is used for post hoc analysis. SNK compares differences from the two groups, which have the largest differences. Additionally, then, it compares the two groups with the second-largest differences. It is powerful to identify the significant difference between group means due to SNK’s stepwise characteristic [35].

## 3. Results and Discussion

### 3.1. Performance Evaluation of the Developed Algorithm

Figure 3 represents the RO performance in constant permeate mode with five different recovery ratios ranging from 38% to 46% in 2% increments (see Appendix A for simulation conditions). The RO system was configured using one pressure vessel with 7 membrane elements. The feed flow rate was set to 250 m^3^/d during the simulated 365 days. The five simulation results showed similar performance trends. The calculated permeate flow rates for each recovery ratio satisfied the target values. The permeate flow rate fluctuations were due to the rate’s upper margin. This was due to a sudden increase in salinity and a drop in seawater temperature below 5 °C in the winter. The operating pressure showed a negative correlation with feed water temperature, decreasing as the feed temperature increased. Relatively low pressure was required to produce the same amount of permeate flow at high temperatures. While the seawater temperature determined the operating pressure trend, the *TDS* concentration determined the local peak of the operating pressure. The local maximum operating pressure was observed when the *TDS* concentration abruptly increased. The temperature and feedwater *TDS* concentration fluctuations had a significant effect on the permeate *TDS* concentration, which followed the feed temperature trend and suddenly increased due to the *TDS* concentration fluctuations in the seawater.

### 3.2. Scenario Study of the RO Configuration 

Figure 4 illustrates the simulation results of the scenario study of each design configuration. The high-pressure pump efficiency for every configuration showed annual mean values of 86% except for design configuration 3 (82%). The median had a higher value than the mean. This means that the high-pressure pump efficiency was mostly closed to the maximum during the one-year simulation period. Although the range of high-pressure pump efficiency was similar to the previously reported study (82–86%), a different result was observed in this study that pump efficiency showed clear differences depending on design configuration [18]. Design configurations 1 and 2 had the same high-pressure pump performance because the number of pumps used in each configuration was 5. However, the increase in the number of pumps from 5 (design configuration 1 and 2) to 6 (design configuration 3) caused a decrease in high-pressure pump efficiency. The fixed number of pumps in pump type 2 (3 pumps) led to the same high-pressure pump efficiency. The median was slightly higher than the mean of the high-pressure pump efficiency for pump type 2. The interquartile range for pump type 1 was narrow than the interquartile range of pump type 2. 

Annual means of *SEC* for every design configuration were under 2 kWh/m^3^, and they had similar interquartile ranges (0.54–0.56 kWh/m^3^), regardless of their pump type. The differences between the *SEC*s’ minimum and maximum were between 1.12 to 1.17 kWh/m^3^. Design configuration 3 showed the most significant difference between minimum and maximum by having 1.17 kWh/m^3,^ and design configuration 4 showed 1.12 kWh/m^3^ as the smallest fluctuation during one year of the simulation period. The difference between design configurations 1 and 3 was the number of pumps. They used the same pump type and consisted of a separated feed line design. Configuration 3 had a higher mean *SEC* value (1.99 kWh/m^3^) than configuration 1 (1.92 kWh/m^3^) because it utilized one more pump than configuration 1. Design configurations 1 and 4 employed the separated feed line design. Although configuration 1 and 4 could produce the same amount of permeate water, configuration 4 only used three pumps due to the capacity of pump type 2. The annual mean *SEC* value for configuration 4 was 1.91 kWh/m^3^, similar to configuration 1. Five trains were used in the design configuration 1 and 6. The difference between configurations 1 and 6 was the design type due to their pump type. Even though two different designs were used for each configuration, they had the same mean *SEC* value.

In the previously reported study, the high-pressure pump efficiency could decrease by approximately 0.3 kWh/m^−3^ when the high-pressure pump efficiency increased from 0.8 to 0.9 [36]. Similarly, Ruiz-García and Nuez [16] reported that the type of membrane could change the *SEC* of process around 0.2 kWh/m^−3^. They also conducted simulations for different feed concentrations. The minimum range of *SEC* was 1.71 to 1.93 and 2.24 to 2.54 kWh m^−3^ for 30,000 and 42,000 mg/L, respectively. Additionally, Jeong et al. [28] showed that an internally staged design could reduce the *SEC* up to 1 kWh/m^−3^ at the designed condition. Compared to the previously reported method for reducing *SEC*, the changes in *SEC* depending on the design configuration for pump type 1 seemed slight decrease. However, the proposed optimization method can further reduce energy consumption when applied with the optimal membrane selection and high-performing pump.

### 3.3. Statistical Analysis Depending on the Design Configuration 

The ANOVA and SNK post hoc analysis were conducted to compare the difference between process performance depending on the design configuration (Table 4). The Levene’s test showed that the p-value of *SEC* was 0.922 (*p* > 0.05). The simulation result of *SEC* for each design configuration had equal variance. The p-value of the ANOVA test for *SEC* was 0.006 (*p* < 0.05), which indicated one of the design configurations had a different annual mean *SEC* value. Table 4 represents the result of SNK post hoc analysis. The *p*-values for configurations 1, 2, 4, 5, and 6 were greater than 0.05. They were no different. However, the *p*-values of design configuration 3 had values between 0.001 and 0.010 (*p* < 0.05). Design configuration 3 had a different annual mean *SEC* value. Every design configuration had a similar mean *SEC* value regardless of pump type and design configuration, except for design configuration 3.

### 3.4. Comparison of the Process Operation Scenarios

Figure 5 illustrates the changes in process performance according to the operation strategy. Every configuration was operable in operation mode 1. However, only three configurations were suitable for operating mode 2 due to the flexibility of the process. The pump efficiency in every configuration dropped by more than 5% when the mode was changed from design mode to operation mode 1. Although the pump efficiency declined, the *SEC* values decreased in all configurations. Configuration 3 especially showed a 1.5% (0.03 kWh/m^3^) reduction in *SEC*, while the pump efficiency decreased by 9%. The considerable decrease in operating pressure prevented a significant increase in specific energy consumption. The reduction in each train’s permeate flow rate led to a *TDS* concentration increase in operation mode 1. Operation mode 2 showed two different trends. The pump efficiency in configuration 2 and 3 decreased by up to 2% when their operation changed from design mode to operation mode 1. The HP pump shutdown minimized the change in feed flow rate per pump. Therefore, the change in operating pressure did not exceed 2 bar and led to a small decrease in the *SEC*. In contrast, in configuration 6, the pump efficiency dropped by 5.53% compared to the initial mode due to the flow rate reduction without a change in the number of pumps. As a result, the *SEC* increased by 0.12 kWh/m^3^ compared to operation mode 1. The permeate *TDS* concentration in operation mode 2 had a similar value as that in design mode. 

## 4. Conclusions

This study was conducted to evaluate the RO process performance according to high-pressure pump and train configuration. One year of monitoring data was analyzed to assess the performance in the expected area of the RO plant installation. An algorithm for RO in constant permeate mode was developed using a previously developed RO model. The effects of design and operation strategy on performance were compared. The following conclusions can be drawn from the results of this study:
(1)The developed RO algorithm showed stable performance under seasonal feedwater temperature and *TDS* concentration variations. The target recovery ratios were successfully reached. The seawater temperature determined the trend, and salinity determined the local peaks of process performance in the Daesan area.(2)The high-pressure pump efficiency and specific energy consumption (*SEC*) for design configuration 3 were significantly different from the others. The deviation of the best operation range led to lower specific energy consumption than others. The ANOVA test result showed pump capacity and pump-train configuration did not significantly affect pump efficiency. The pump’s operation in the best operation range only decided the process performance, not the pump capacity. (3)Between the two operation strategies, operation strategy 1 was more efficient than operation strategy 2. The reduction in operating pressure mitigated the decrease in HP pump efficiency. The process flexibility was decided depending on the number of pumps and trains. Hence, design configuration 2, 3, and 6 were operable in two different operation strategies. (4)The process performance was mainly compared between cases in terms of *SEC*. Therefore, further research is required to optimize the process with respect to capital and operational expenditure. 

## Figures and Tables

**Figure 1 membranes-12-01145-f001:**
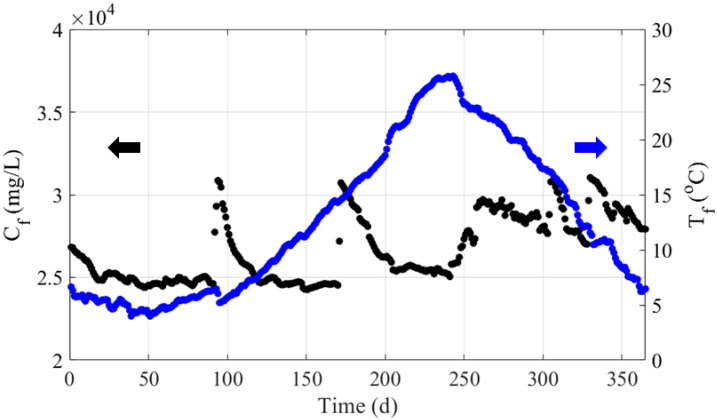
Graphical description of seawater temperature and salinity offshore from Daesan during the studied 365 days of monitoring.

**Figure 2 membranes-12-01145-f002:**
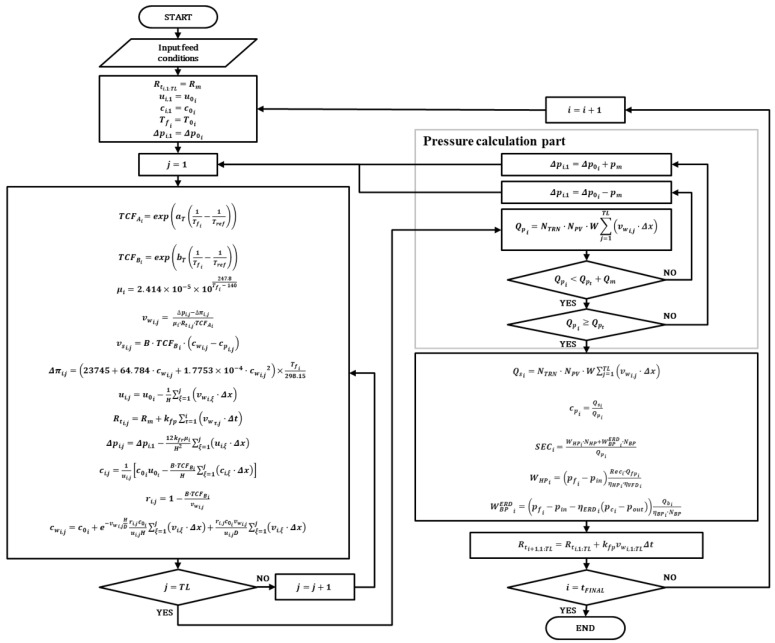
Flow chart for the seawater reverse osmosis process model in constant permeate mode with time-variant feed water condition input.

**Figure 3 membranes-12-01145-f003:**
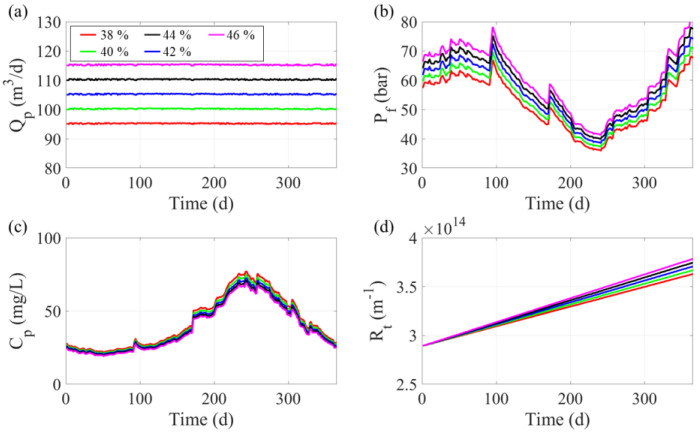
Simulation results for the seawater reverse osmosis algorithm with respect to five different recovery ratios: (**a**) permeate flow rate (m^3^/d), (**b**) operating pressure (bar), (**c**) permeate total dissolved solids concentration (mg/L), and (**d**) total membrane resistance (m^−1^).

**Figure 4 membranes-12-01145-f004:**
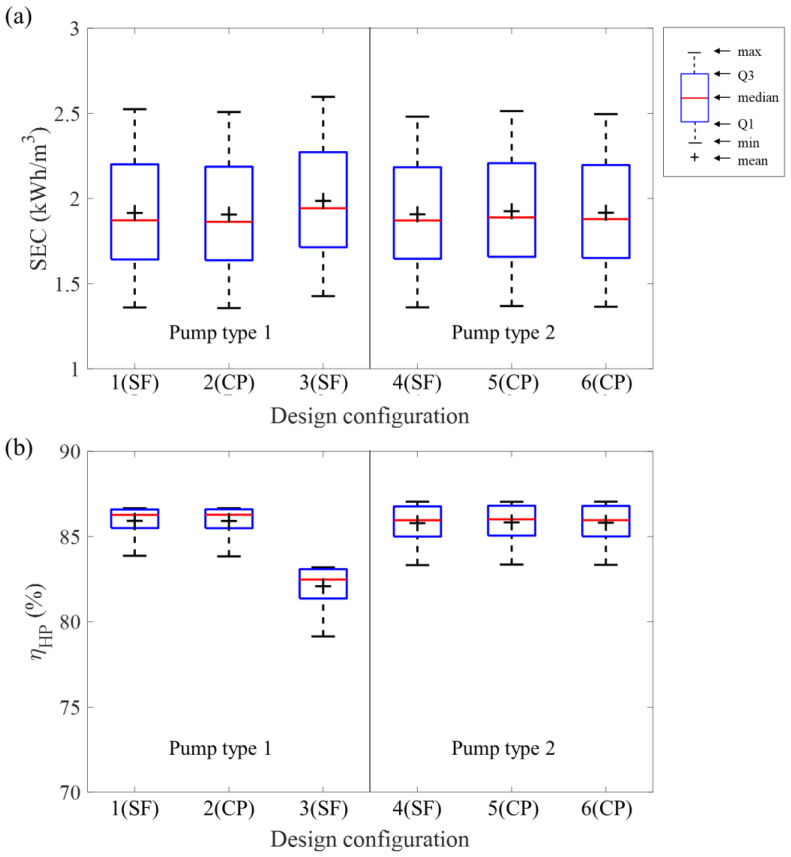
Simulation results depending on the design configuration: the annual statistics for (**a**) high-pressure pump efficiency, (**b**) specific energy consumption. The design configuration from 1 to 3 and design configuration from 4 to 6 are configured with pump type 1 and pump type 2, respectively. It is noted that the Q1 and Q3 in the legend of figure indicate the first and third quartiles, respectively.

**Figure 5 membranes-12-01145-f005:**
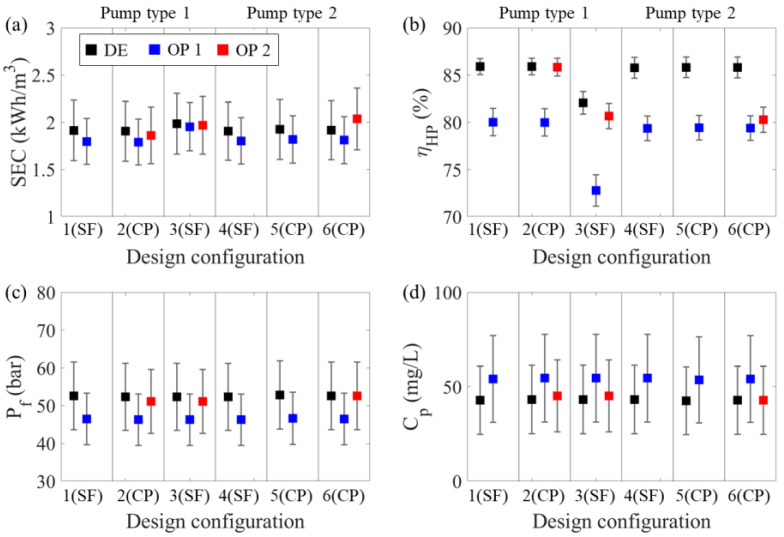
Operating performance according to design configuration and operation strategy: the annual statistics for (**a**) specific energy consumption (kWh/m^3^), (**b**) pump efficiency, (**c**) operating pressure (bar), and (**d**) permeate *TDS* concentration (mg/L). The squares indicate the mean of each value. The error bars denote standard deviations above and below the mean values.

**Table 1 membranes-12-01145-t001:** Parameter values for the process simulation.

Parameter	Value
SWRO model operating conditions	
VFD ^a^ efficiency $\eta_{VFD}$ (%)	95
Booster pump efficiency $\eta_{BP}$(%)	85
ERD ^b^ efficiency $\eta_{ERD}$(%)	95
Operation duration (days)	365
Membrane element properties (for 8-in. RO element)	
Intrinsic membrane resistance *R_m_* (m^−1^)	2.89 × 10^14^
Salt permeability coefficient *B* (m/s)	9.36 × 10^−9^
Spacer thickness *H* (m)	8.64 × 10^−4^
Membrane channel width *w* (m)	37
Membrane channel length *L* (m)	1
Number of membrane elements in a pressure vessel	7
Hydrodynamic properties	
Fouling potential of feedwater, *k_fp_* (m^−2^)	5.5 × 10^11^
Hydraulic dispersion coefficient *D* (−)	9.6 × 10^−9^
Friction coefficient due to the membrane spacer *k_fr_* (−)	8

^a^ Variable frequency drive. ^b^ Energy recovery device.

**Table 2 membranes-12-01145-t002:** Coefficients for the high-pressure pumps according to pump type. The coefficient of determination (*R*^2^) was 0.99 for both types of pump.

Coefficients	Pump Type	
	1	2
*p* _1_	−32.75	−26.51
*p* _2_	0.3594	0.2085
*p* _3_	0.8518	0.6227
*p* _4_	−0.0003163	−0.0001112
*p* _5_	−0.0004268	−0.0001544
*p* _6_	−0.005476	−0.003594

**Table 3 membranes-12-01145-t003:** Design and operation configuration employed during the optimization and simulation of reverse osmosis for a single-pass SWRO system.

	Permeate Flow Rate (m^3^/d)	Design Configuration	Pump Type	Design Type ^a^	Pumps	Trains	PVs ^b^
Designed configuration	60,000	1	1	SF	5	5	625
		2	1	CP	5	6	630
		3	1	SF	6	6	630
		4	2	SF	3	3	630
		5	2	CP	3	4	620
		6	2	CP	3	5	625
Operation strategy 1	48,000	1	1	SF	5	5	625
		2	1	CP	5	6	630
		3	1	SF	6	6	630
		4	2	SF	3	3	630
		5	2	CP	3	4	620
		6	2	CP	3	5	625
Operation strategy 2	48,000	2	1	CP	4	5	525
		3	1	SF	5	5	525
		6	2	CP	3	4	500

^a^ SF and CP indicate the pump-train configuration with separated feed lines and common pressure center design, respectively. ^b^ PV denotes the pressure vessel.

**Table 4 membranes-12-01145-t004:** Result of SNK post hoc analysis for the specific energy consumption depending on the design configuration.

*p*-Value		Design Configuration				
		2	3	4	5	6
Design configuration	1	0.919	0.014	0.746	0.900	0.959
	2		0.009	0.947	0.922	0.971
	3			0.007	0.010	0.009
	4				0.872	0.925
	5					0.700

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
