# Peer review of "Optimization of the Design Configuration and Operation Strategy of Single-Pass Seawater Reverse Osmosis"

_membranes, 2022, doi:10.3390/membranes12111145_

Round 1

Reviewer 1 Report

The manuscript is very useful for the long-time operation of the RO plant. The manuscript  can be published in Membranes after addressing the following points.

1) A little more background information is needed, i.e. optimal design and operation of RO system. The related references should be cited.

2) A figure of the process flow in Table 3 should be given.

3) A detailed description of the legend in figure 4 is needed. The meanings of Q1 and Q3 are not clear.

4) Figure 3 gives the RO performance in constant permeate mode with different recovery ratios, however, the system design configuration for the case is not given.

5) Constant values are assumed for the efficiencies of VFD and boost pumps. Actually, these efficiencies are changed with time during operation. A discussion should be given.

Reviewer 2 Report

The review manuscript titled “Optimization of the design configuration and operation strategy of single-pass seawater reverse osmosis” and written by Seung Ji Lim et al. is interesting and reports a performance analysis of a SWRO system considering different configurations and the variation of feed water temperature and concentration during a year. The paper has some drawbacks that should be fixed by the authors before being considered for publication. I recommend a major revision based on the following comments:

1.       The introduction does not cover enough relevant studies related with the optimization of SWRO systems in terms of configuration and operation. Without the consideration of the mentioned studies, the novelty cannot be proved, and the results cannot be compared with those obtained in other studies. Please, extend the section introduction. (Membranes 2021, 11(11), 906; Desalination 533, 115768; Computers & Chemical Engineering 153, 107441; Membranes 2021, 11(4), 243; Membranes 2021, 11(2), 138)

2.       SEC and TDS are variables, please, write them in italics.

3.       From where the authors got the data of Table 1?. The efficiency of the booster performance it seems very high considering that the performances of these devices, for example for ERI is around 65%.

4.       In page 8, Table 3 shows different operating conditions in terms of permeate flow, pump type, number of pumps and trains. These considerations are quite specific, so the authors should specify in the title. For example, “…a single-pass SWRO system”.

5.       Did the authors consider the pressure drop increase due to membrane fouling? (IFAC-PapersOnLine 54 (3), 158-163) Fouling does not only impact on permeability coefficients but also on the pressure drop.

6.       The SEC strongly depends on feedwater temperature. I guess the results of Figure 4 are in terms of Annual average right? Please, be specific in the caption of the figure.

7.       Usually, the performance of HPP is provided by the pump manufacturer. From where the authors got the Equation 21? Its validity must be demonstrated as well as the data of Table 2.

8.       To evaluate the performance of the system, in terms of SEC, the authors have supposed different pumps and different feed flows by changing the number pressure vessels. RO system performance considering a wide operation conditions including feed flow have been studied by other authors, please, compare the obtained results

Round 2

Reviewer 2 Report

The authors have addressed all my comments